# Frequency-specific neural signatures of perceptual content and perceptual stability

**Richard Hardstone[1], Matthew W Flounders[1], Michael Zhu[1], Biyu J He[1,2,3,4]\***

[1]Neuroscience Institute, New York University Grossman School of Medicine, New York, United States; [2]Department of Neurology, New York University Grossman School of Medicine, New York, United States; [3]Department of Neuroscience and Physiology, New York University Grossman School of Medicine, New York, United States; [4]Department of Radiology, New York University Grossman School of Medicine, New York, United States

**Abstract** In the natural environment, we often form stable perceptual experiences from ambiguous and fleeting sensory inputs. Which neural activity underlies the content of perception and which neural activity supports perceptual stability remains an open question. We used a bistable perception paradigm involving ambiguous images to behaviorally dissociate perceptual content from perceptual stability, and magnetoencephalography to measure whole-brain neural dynamics in humans. Combining multivariate decoding and neural state-space analyses, we found frequency-band-specific neural signatures that underlie the content of perception and promote perceptual stability, respectively. Across different types of images, non-oscillatory neural activity in the slow cortical potential (<5 Hz) range supported the content of perception. Perceptual stability was additionally influenced by the amplitude of alpha and beta oscillations. In addition, neural activity underlying perceptual memory, which supports perceptual stability when sensory input is temporally removed from view, also encodes elapsed time. Together, these results reveal distinct neural mechanisms that support the content versus stability of visual perception.

**\*For correspondence:**
biyu.he@nyulangone.org

**Competing interest:** The authors declare that no competing interests exist.

## Editor's evaluation

Bistable visual perception offers a unique window to study how perception arises and changes via an interaction between bottom-up and top-down processes. In three Magnetoencephalography (MEG) experiments with advanced neural state space analysis, this study demonstrates that two key aspects of bistable visual perception – perceptual content and perceptual stability – are mediated by slow cortical potential (SCP) and α-β-band neural oscillations, respectively. The findings will be of interest for many fields, including those studying perception, consciousness, and attention.

## Introduction

How vivid visual perceptual experiences are generated by the brain remains a central question in neuroscience. There are two critical functions that the visual perceptual system is able to accomplish: the first is to generate the specific content of perceptual experience (such as seeing a predator); the second is to maintain a stable perceptual experience despite noisy and unstable retinal input due to constant head and eye movements and complexities of the natural environment involving occlusion, shading, and dynamic changes of sensory input (e.g., the predator is camouflaged and hidden in

the bush). Here, we investigate neural mechanisms giving rise to specific perceptual content and supporting perceptual stability in the human brain.

Motivated by several strands of previous work, we hypothesized that there might be different components of electrophysiological neural activities that support perceptual content and perceptual stability, respectively. First, previous studies using multivariate analysis to decode perceptual content based on electroencephalography/magnetoencephalography (EEG/MEG) activity have typically reported greater successes when the decoder's input was raw filtered field potentials in the relatively low (<30 Hz) frequency range (e.g., *Carlson et al., 2013*; *Salti et al., 2015*; *King et al., 2016*). Further, studies using frequency-band-specific analysis have shown that the ability to decode perceptual content is contributed most by the slow cortical potential (SCP, <5 Hz) frequency range (*Baria et al., 2017*; *Flounders et al., 2019*). This is consistent with the $1/f$ distribution of EEG/MEG power spectrum suggesting that the SCP band contributes most to the power in the event-related potential/field (ERP/ERF) frequency range (*He et al., 2010*; *He, 2014*; *Donoghue et al., 2020*).

Second, a line of work focused on brain oscillations has shown that moment-to-moment fluctuations of alpha oscillation amplitude in sensory cortices modulate local cortical excitability in a manner that transcends specific stimulus/perceptual contents (*Samaha et al., 2020*). In addition, alpha and beta oscillations can carry top-down feedback influences (*van Kerkoerle et al., 2014*; *Michalareas et al., 2016*), and top-down feedback may facilitate resolving perceptual ambiguity by carrying information consistent with prior knowledge (*Cavanagh, 1991*; *Yuille and Kersten, 2006*). We therefore hypothesized that there might exist a frequency-band separation between neural activity supporting

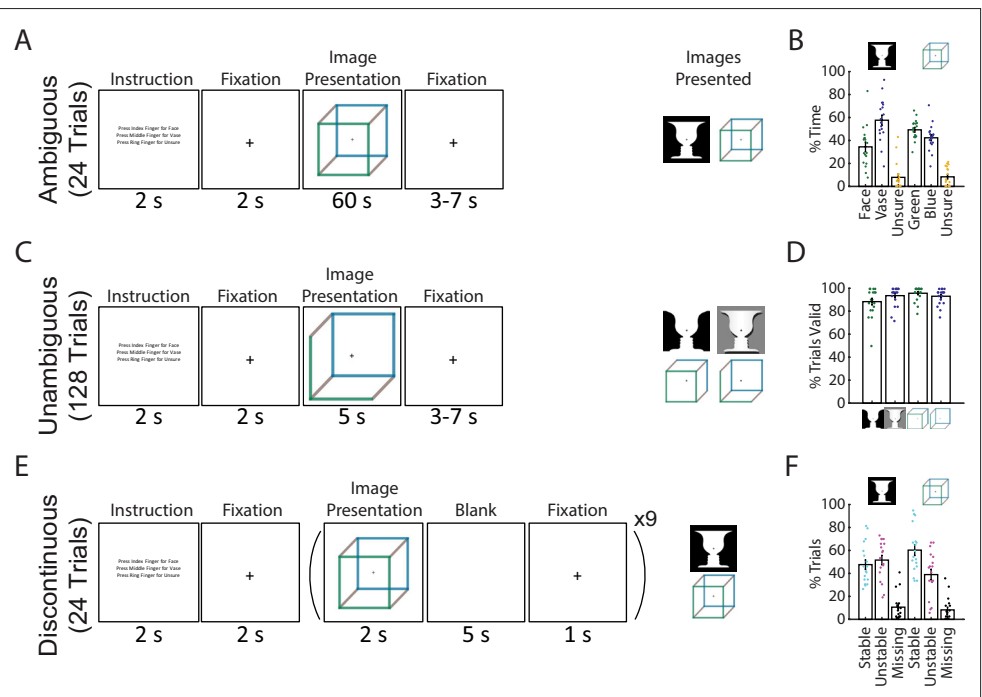

**Figure 1.** Paradigm and behavioral results. (**A**) In the *Ambiguous* condition, bistable images were presented for 60 s each, and subjects pressed buttons to indicate their current percept. (**B**) Percentage of time spent in each perceptual state during the *Ambiguous* condition. (**C**) In the *Unambiguous* condition, bistable images which were modified to reduce their ambiguity were presented. (**D**) Percentage of valid trials for each image type during the Unambiguous condition. Valid trials consisted of the subject pressing the button only once for the intended percept. (**E**) In the *Discontinuous* condition, ambiguous images were shown nine times with interleaving blank periods. (**F**) Percentage of blank periods that were classified as Stable, Unstable, or Missing. 'Stable' indicates that perception was the same before and after the blank period, 'Unstable' that it was different, and 'missing' indicates that no button press was recorded during the pre or post image presentation period. (**B, D, F**) Dots indicate individual subjects; bars and error bars indicate group mean and standard error of the mean (SEM).

The online version of this article includes the following figure supplement(s) for figure 1:

**Figure supplement 1.** Individual-level power spectra and group-level duration histograms.

perceptual content and neural activity supporting perceptual stability, with the former residing in the non-oscillatory activity in the SCP range, and the latter predominantly residing in oscillatory activity in the alpha/beta range.

To test this hypothesis, we recorded whole-head MEG while participants performed a bistable visual perception task involving two different ambiguous figures (Necker cube and Rubin face–vase illusion). Data from these two images were separately analyzed, providing a within-study reproducibility and generalizability check. In two different conditions, the images were either continuously presented (*Ambiguous* condition, *Figure 1A*) or intermittently presented (*Discontinuous* condition, *Figure 1E*). The *Ambiguous* condition allowed us to dissociate perceptual content (perceiving one or the other interpretation) from perceptual stability (how long a percept lasts). The *Discontinuous* condition allowed us to investigate the neural underpinnings of perceptual memory: previous research has shown that perceptual alternations slow down during intermittent presentation and that a perceptual memory trace exists in the intervening blank periods such that the recently experienced percept is likely reinstated when the image reappears (*Orbach et al., 1966*; *Leopold et al., 2002*; *Pearson and Brascamp, 2008*). This phenomenon provides a window into neural mechanisms supporting perceptual stability when sensory input is both ambiguous and fleeting, as often is the case in natural vision. Finally, to test the generalizability of the identified neural correlate of perceptual content, participants additionally performed a task in which modified versions of the Necker cube and Rubin face–vase images with ambiguity removed (*Wang et al., 2013*) were presented and perceptual content varied with the actual physical stimulus (*Unambiguous* condition, *Figure 1C*).

To test our hypothesis, we combined multivariate decoding with an innovative multivariate regression approach which allowed us to identify separate neural subspaces relevant to the encoding of different types of behavioral information that are simultaneously present in the same task (*Mante et al., 2013*)—specifically, perceptual content and perceptual switching dynamics in the present task. Across two different task conditions with different levels of stimulus ambiguity (*Ambiguous* and *Unambiguous*), we found that non-oscillatory neural activity in the SCP range, but not alpha or beta oscillations, encoded perceptual content. Furthermore, across both *Ambiguous* and *Discontinuous* conditions, we found that the fluctuations of alpha and beta amplitudes modulated perceptual stability and perceptual memory. Interestingly, we also found that SCP modulated perceptual stability, although with less spatial consistency across subjects than alpha and beta oscillations. These results reveal an intriguing frequency-domain separation of neural activity encoding perceptual content and that supporting perceptual stability.

## Results

### Task paradigm and behavioral results

We recorded 18 subjects with whole-head MEG (CTF, 272 functional axial gradiometers) performing 3 conditions of a visual perception task involving 2 commonly studied ambiguous figures (Necker cube and Rubin face–vase illusion). The first condition consists of the classic bistable perception task (*Ambiguous*, *Figure 1A*), in which subjects viewed ambiguous images for 60 s at a time and used button presses to indicate their spontaneous perceptual switches (with three buttons corresponding to two of the possible percepts and an 'unsure' option). In the second condition (*Unambiguous*, *Figure 1C*), subjects viewed modified versions of these images for 5 s at a time, which enhance one of the possible interpretations and largely abolish perceptual switching (*Wang et al., 2013*); subjects indicated their percepts in a similar fashion as before. In the final condition (*Discontinuous*, *Figure 1E*), each ambiguous figure was presented repeatedly with interleaving blank periods, allowing us to investigate neural underpinnings of perceptual memory during the blank periods (*Leopold et al., 2002*; *Pearson and Brascamp, 2008*); subjects indicated their percepts whenever the image was in view.

During the *Ambiguous* condition (*Figure 1A, B*) perceptual switching occurred, with group-level results showing that each of the possible percepts was perceived (on average >25% of the time), and that subjects were rarely unsure of which percept they were experiencing (<10% occurrence, these time periods were removed from further analyses). Modifying the images to be unambiguous (*Figure 1C, D*) was successful, as evidenced by subjects having on average >80% valid trials (defined as trials with only one button press indicating the intended percept). In the *Discontinuous* condition, we found an increased likelihood that perception remained stable across the blank period for the

Necker cube (one-tailed Wilcoxon sign-rank (17) = 116.5, p = 0.031) but not for the Rubin face–vase image (one-tailed Wilcoxon sign-rank (17) = 71.5, p = 0.736).

Together, these behavioral results demonstrate the classic bistable perception phenomenon in the *Ambiguous* condition, successful disambiguation of the images in the *Unambiguous* condition, and a means to investigate perceptual memory by contrasting stable- and unstable-blank periods in the *Discontinuous* condition. Importantly, the use of two different ambiguous figures in all three conditions allowed us to test whether the neural findings are reproducible and generalizable across the specific stimulus characteristics. Here, taking advantage of the large-scale neural dynamics recorded by whole-head MEG, we aimed to dissociate dynamical neural activity underlying perceptual content and supporting perceptual stability, respectively.

## Perceptual content can be decoded from SCP but not amplitude of band-limited oscillations

In the classic bistable perception task, perceptual content experienced by the subject continuously alternates between two possible outcomes while the sensory input stays constant. This allows the investigation of the neural correlates of the content of conscious perception while controlling for low-level sensory processing. To this end, we applied time-resolved multivariate decoding to whole-brain MEG data (for details, see *Methods*). We tested three components of neural field potentials—SCP (<5 Hz), alpha-band amplitude (amplitude envelope of 8–13 Hz filtered data) and beta-band amplitude (amplitude envelope of 13–30 Hz filtered data)—in their ability to distinguish between the two percepts that are alternatively experienced for each ambiguous figure. The SCP activity corresponds to the low-frequency component of the broadband, non-oscillatory (i.e., aperiodic) activity (*He et al., 2010*; *He, 2014*), while the alpha and beta bands have prominent oscillatory activity (*Figure 1—figure supplement 1A, B*). After extracting the relevant neural feature, perceptual content decoding was performed using a fourfold cross-validated linear support vector machine (SVM), with significance determined using cluster-based permutation testing that corrects for multiple comparisons across time.

To investigate neural activity underlying specific perceptual content, we selected time periods ('subtrials') that were preceded and followed by button presses for two different percepts (i.e., excluding periods preceded or followed by unsure presses, or at the beginning or end of the image presentation) and sorted them into two groups. Thus, each subtrial begins with a button press indicating the relevant percept and ends with a button press that indicates a switch to the opposite percept. These subtrials were of different lengths (*Figure 1—figure supplement 1C, D*), as percept duration is highly variable during spontaneous bistable perception—a topic we will address in the following section. To decode perceptual content, we then subsampled each subtrial by taking 100 equally spaced time points from the beginning to the end of that subtrial (henceforth referred to as percentile of a percept). This way, we tested whether a decoder trained using neural features recorded at the beginning (/middle/end) of a subtrial generalized to the beginning (/middle/end) of other subtrials, even if they were of different lengths (*Figure 2A*).

Decoding accuracies over the course of a percept for the three different neural features are shown in *Figure 2C*, left column. Significant decoding of perceptual content is found for SCP, but not alpha or beta amplitude (except for a very small temporal cluster for the face–vase image), and SCP shows significantly stronger decoding than alpha or beta amplitude (*Figure 2—figure supplement 1*) suggesting that the currently experienced percept is most strongly encoded in SCP activity. These findings are consistent with previous results using other visual perceptual tasks (*Baria et al., 2017*; *Flounders et al., 2019*). To shed light on whether the neural representation of the percept stays stable over the duration of the percept or changes constantly over time, we tested the temporal generalization of the decoder, whereby decoders trained at each time point are tested at all other time points (*King and Dehaene, 2014*). We observed broad decoder temporal generalization in the SCP band for both the Necker cube and Rubin face–vase illusion (*Figure 2C*, right column), especially from 20th to 80th percentile of the percept duration. This suggests that neural representation underlying perceptual content, except at the very beginning and end of a percept, is relatively stable over time regardless of percept duration, and localizes to the SCP band in the frequency domain.

We next tested whether a similar pattern of findings exists when stimulus ambiguity is removed, by decoding perceptual content using data from the *Unambiguous* condition. To this end, we

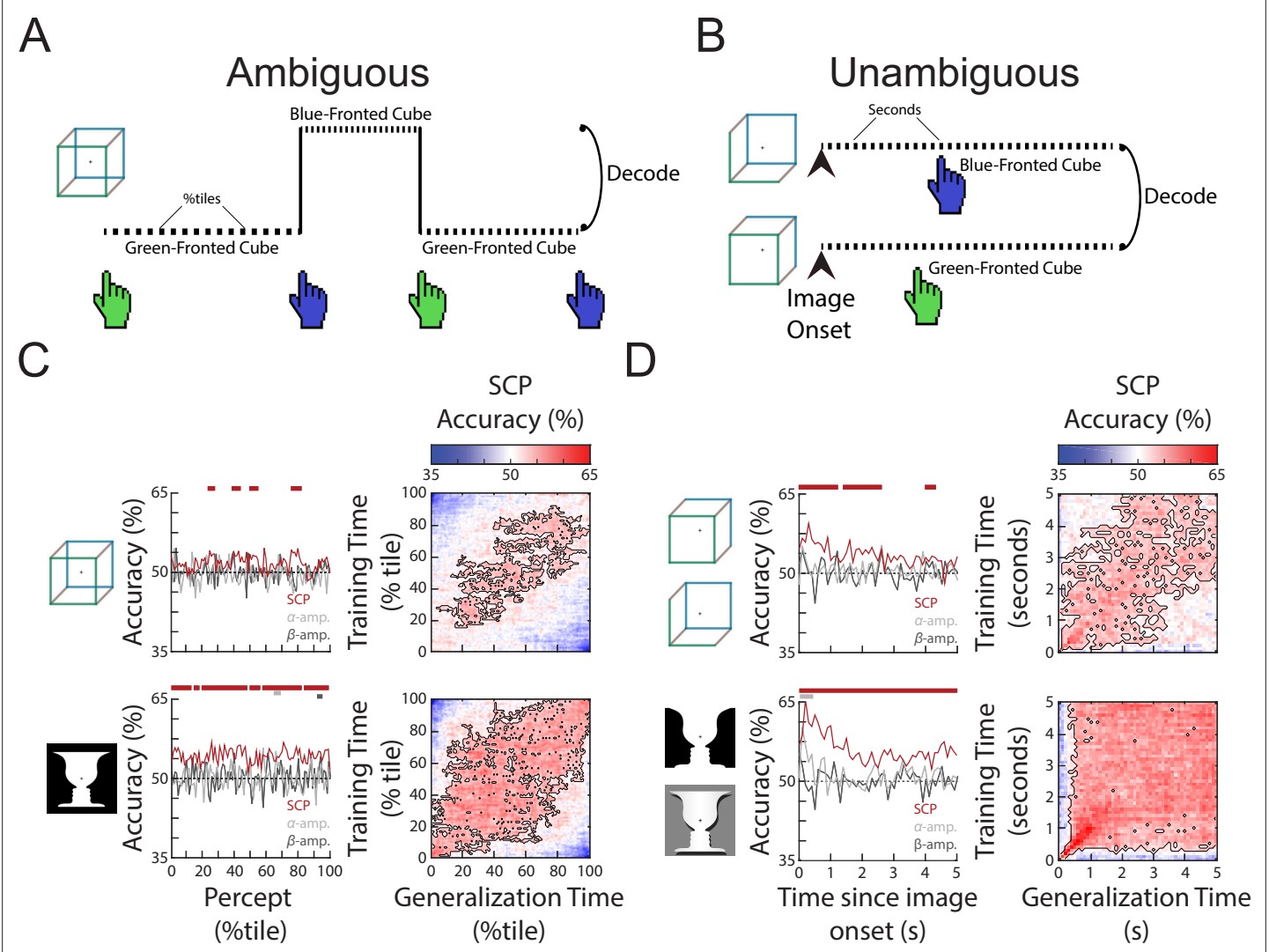

**Figure 2.** The content of subjective perception is encoded in slow cortical potential (SCP) activity. (**A**) In the *Ambiguous* condition, subjects continuously reported their current perception using button presses. Neural activity during each percept was split into 100 percentiles according to time elapsed, and percept was then decoded at each temporal percentile. (**B**) In the *Unambiguous* condition, different disambiguated images were shown that emphasized one of the two percepts. For valid trials in which subjects' reported percept matched the intended percept (see *Figure 1D*), image content was decoded separately at each time point during image presentation. (**C**) (*Left*) Decoding accuracy for perceptual content during the time course of a percept. Significant temporal clusters of percept decoding exist for both images using SCP as the decoder input, but not when alpha or beta amplitude was used as decoder input. (*Right*) Temporal generalization matrices. Significant clusters are outlined, showing generalization across a large proportion of the percept duration. (**D**) (*Left*) Decoding accuracy for unambiguous images during their presentation. Significant temporal clusters of image/percept decoding exist in the SCP range throughout image presentation, but not for alpha/beta amplitude. (*Right*) Temporal generalization matrices showing significant generalization across a large proportion of the image presentation duration.

The online version of this article includes the following figure supplement(s) for figure 2:

**Figure supplement 1.** Slow cortical potential (SCP) has significantly higher decoding than alpha or beta oscillations in both the ambiguous (top) and unambiguous (bottom) conditions.

selected valid trials (wherein the subject only had one button press indicating the intended percept), which account for the vast majority of all trials (*Figure 1D*), and constructed decoders to distinguish between the two different perceptual contents which coincided with different image inputs (i.e., decoding between the two versions of face–vase image, and between the two versions of cube image, *Figure 2B*). Similar to the *Ambiguous* condition, significant perceptual content decoding was obtained using SCP activity, but not alpha or beta amplitude (except for one small temporal cluster at image onset for alpha amplitude, face–vase image) (*Figure 2D*, left column) and decoding was

stronger in the SCP band (*Figure 2—figure supplement 1*). Decoding accuracy in the SCP band was highest in the first second after image onset and then drops to a lower level (likely due to neural adaptation). The higher decoding accuracy in the *Unambiguous* condition as compared to the *Ambiguous* condition is likely due to the differences in sensory input that coincides with different perceptual contents, as well as consistent timing across all trials (all image presentations last 5 s, as opposed to variable percept durations in the *Ambiguous* condition). Lastly, as in the *Ambiguous* condition, the SCP decoder of perceptual content generalized well across time in the *Unambiguous* condition (*Figure 2D*, right column), suggesting that the underlying neural code is stable over time after the very initial image onset-related activity.

Together, these results show that perceptual content information is decodable from SCP activity, but not from the amplitude of alpha or beta oscillations, regardless of whether sensory input is ambiguous or not. In the next section, we investigate neural processes controlling the stability of a percept as compared to the neural processes underlying the content of that percept.

### Defining a behaviorally relevant neural subspace

To simultaneously extract neural activity relevant to different behavioral metrics—here, the content of perception and the dynamics of perceptual switching—we adapted a multivariate analysis approach recently developed in animal neurophysiology ('state-space analysis') (*Sussillo, 2014*). In this framework, multivariate neural activity (across neurons or sensors) at each time point corresponds to a

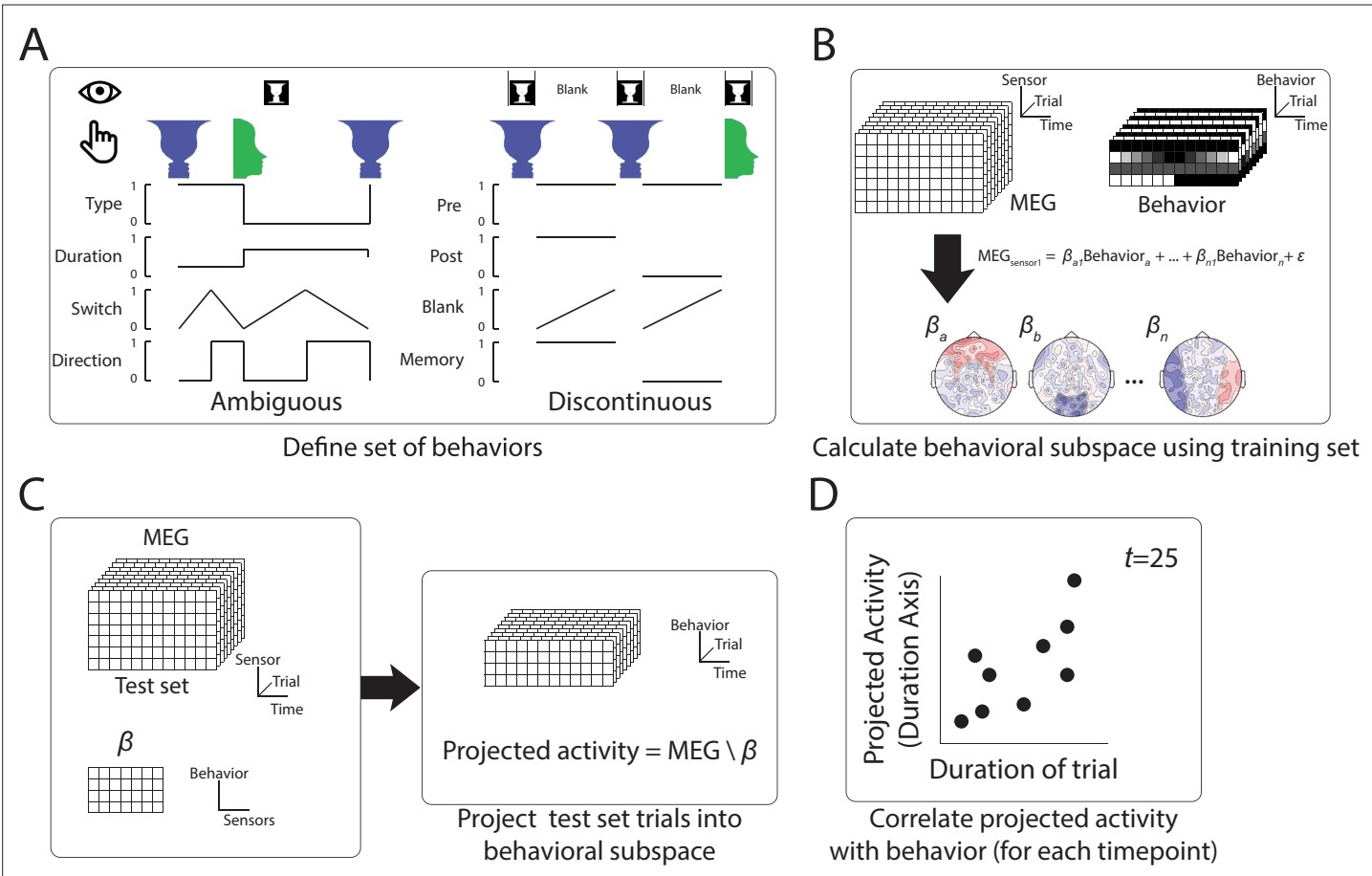

**Figure 3.** Method to define behaviorally relevant neural subspace. (**A**) For the *Ambiguous* and *Discontinuous* conditions, a set of behavioral metrics was defined, which incorporated information about both the perceptual content and the perceptual switching dynamics. (**B**) Using a training set of MEG data, neural subspaces were defined by apply multilinear regression to activity from each sensor using the four behavioral metrics as predictors. The weights across sensors for each behavioral metric define the neural subspace relevant to that metric. (**C**) The test MEG dataset can then be projected into the subspace corresponding to a particular behavioral metric to provide a prediction of the metric value at each time point. (**D**) Accuracy of the prediction can then be tested by comparison to the actual behavioral data.

specific location in the neural state space, where each dimension is a neuron/sensor. Because different neurons/sensors are highly correlated and not all are informative for the behavior of interest, dimensionality reduction methods (such as principal component analysis, PCA) are typically applied to identify a low-dimensional subspace capturing the majority of the variance in the data and/or most relevant to the behavior in question (*Briggman et al., 2005*; *Churchland et al., 2012*; *Stokes et al., 2013*; *Baria et al., 2017*). Here, following earlier studies (*Mante et al., 2013*; *Kayser et al., 2016*), we identify the neural subspace most relevant to a particular behavioral metric by conducting a multilinear regression using orthogonal, task-related axes that capture perceptual content and perceptual switching dynamics, as described in detail below. Importantly, unlike the decoding approach employed in the earlier analysis, where a different decoder is trained for each time point within a trial and decoder weights are sometimes difficult to interpret (*Haufe et al., 2014*), the state-space analysis aims to identify a neural subspace that is unchanging across time, wherein the trajectory of neural activity informs about changes in behavior across time.

For both the *Ambiguous* and *Discontinuous* conditions, we defined a set of behavioral axes capturing both perceptual content and perceptual switching dynamics (*Figure 3A*). For the *Ambiguous* condition, these consisted of a *Type* Axis, which was a binary (0 or 1) variable indicating the current perceptual content; a *Duration* Axis, indicating the overall duration of the current percept; a *Switch* Axis, indicating the temporal distance to a reported perceptual switch (i.e., button press); and, finally, a *Direction* Axis, a binary variable indicating whether the current percept is stabilizing or destabilizing (operationalized as the first half vs. second half of a percept). Both the Switch and Duration axes had values normalized to the range of [0, 1], such that for the *Switch* axis, time points corresponding to button presses are 0 and time points furthest away from button presses within each percept are 1; for the *Duration* axis, the shorted and longest percept durations for a particular subject are coded as 0 and 1, respectively. Thus, the *Switch*, *Duration*, and *Direction* axes together capture different aspects of the perceptual switching dynamic, while the *Type* axis captures the specific perceptual content.

For the *Discontinuous* condition, because we are interested in neural activity underlying the perceptual memory trace, only blank periods were analyzed. Two behavioral axes were defined to capture the content of perception: *Pre* and *Post*, which are binary variables indicating the perceptual content before and after the blank period, respectively. Two behavioral axes were defined to capture the dynamics of perceptual memory: a *Memory* axis, which is a binary variable indicating whether a perceptual memory trace is present or absent (defined by whether the reported perceptual content before and after the blank period is the same); and a *Blank* axis, which is a linearly ramping variable between 0 and 1 from the beginning to the end of each blank period, indicating how long the perceptual memory trace, if present, has last since image offset.

To extract the neural subspace most relevant to each behavioral metric, half of the data were used as a training set and a multilinear regression model was fit for each sensor (*Figure 3B*). The $\beta$ weights from the regression model provide an estimate of the relative contributions of the activity of that sensor to each of the different behavioral metrics. The set of weights across sensors for a particular behavioral metric thus defines the identified neural subspace. The held-out test data are then projected into that subspace (*Figure 3C*), which provides a prediction of the value of that behavioral metric at each time point, solely based on the MEG activity. Comparing the actual and predicted behavior (*Figure 3D*) yields a cross-validated estimate of how much information the identified neural subspace has about that aspect of behavior.

Lastly, the set of $\beta$ weights (with size equal to the number of MEG sensors) for each behavioral metric can also be inspected for consistency at the group level (topography in *Figure 3B*) to determine whether neural activity from a particular sensor significantly contributes to a particular behavioral metric. In sum, this state-space analysis allows us to simultaneously identify neural underpinnings of multiple aspects of behavior at once.

## Role of neural oscillations in perceptual switching dynamics during bistable perception

We first applied the state-space analysis to data from the *Ambiguous* condition. For perceptual content ('Type' axis), we found that the neural subspace identified using the SCP activity allows robust prediction of moment-to-moment perceptual content experienced by the subject for the face–vase image (*Figure 4—figure supplement 1B*, 'Type' column), with highly consistent sensor-level weights across

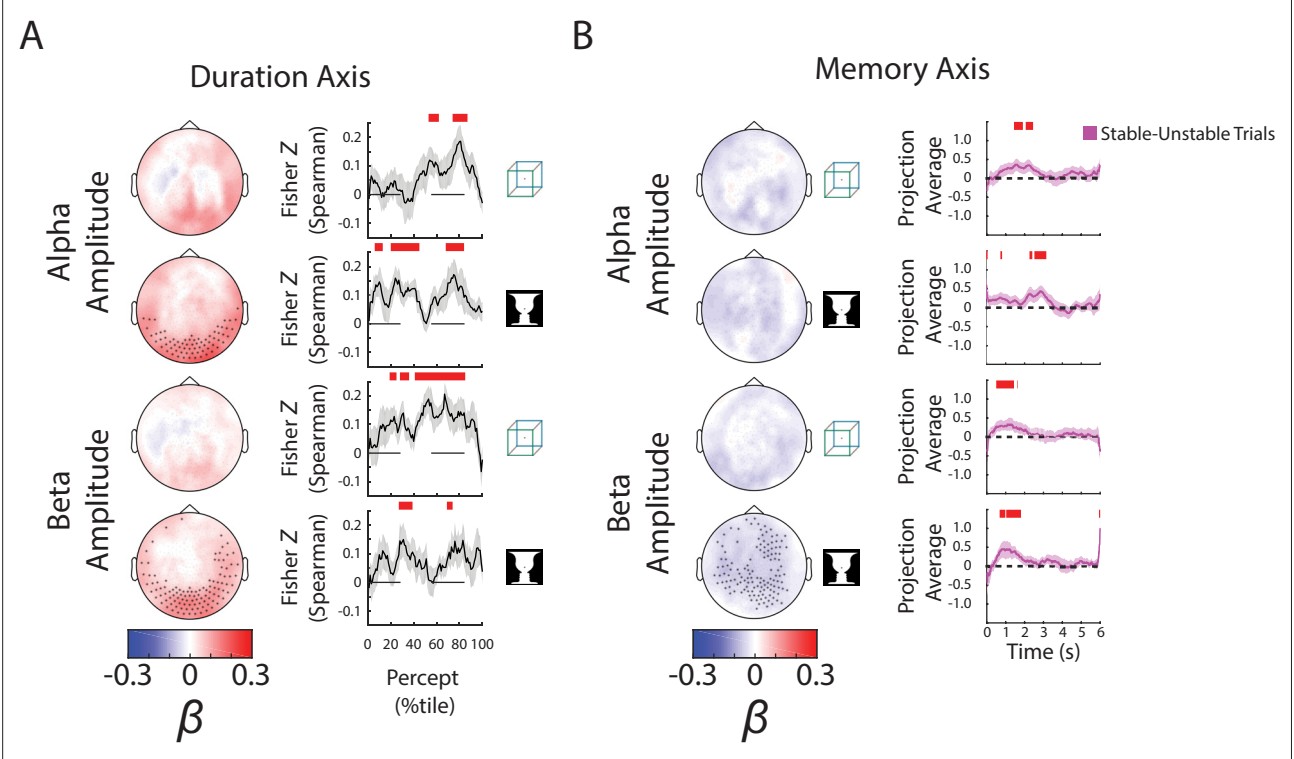

**Figure 4.** Alpha and beta amplitudes are involved in the switching process and perceptual memory trace. (**A**) (*Left*) Group-average $\beta$ weights for Duration axis in the ambiguous condition. Sensors whose $\beta$ weights are significantly different from zero (Wilcoxon signed-rank test, cluster-corrected, p < 0.05) are marked with *x*. (*Right*) Mean Fisher *z*-transformed Spearman rho values, obtained by correlating predicted and actual behavioral values across trials for each subject. Shaded areas show group-level standard error of mean (SEM). Significant time points (p < 0.05, cluster-based permutation test) are indicated by the horizontal red bars. (**B**) (*Left*) Group-average $\beta$ weights for Memory axis in the discontinuous condition. Sensors whose group-level $\beta$ weights are significantly different from zero (Wilcoxon signed-rank test, cluster-corrected, p < 0.05) are marked with *x*. (*Right*) Difference in neural activity projected onto the Memory axis between the Stable and Unstable trials (i.e., blank periods sandwiched by the same percept or different percepts). Significant differences (p < 0.05, cluster-based permutation test) between them are indicated by red horizontal bars.

The online version of this article includes the following figure supplement(s) for figure 4:

**Figure supplement 1.** Complete neural subspace results for the *Ambiguous* condition.

**Figure supplement 2.** Complete neural subspace results for the *Discontinuous* condition.

subjects (*Figure 4—figure supplement 1A*, 'Type' column). Interestingly, the results did not reach significance for the cube image, consistent with weaker perceptual content decoding for the cube than face–vase image (*Figure 2A*) and potentially due to the decoder being retrained at each time point whereas the subspace is fixed across time points. Using amplitude of alpha or beta oscillations, we could not achieve significant prediction of perceptual content (except for a small temporal cluster for beta amplitude, cube image) (*Figure 4—figure supplement 1B*, 'Type' column) and there were no consistent weights across subjects (*Figure 4—figure supplement 1A*, 'Type' column). Together, these results reinforce the impression from the decoding results showing that perceptual content information largely localizes to the SCP band, manifesting as moment-to-moment changes in large-scale SCP activity.

Focusing on the 'Duration' axis, which captures variability in the percept durations, we found consistent group-level $\beta$ weights in occipital cortex for alpha and beta amplitudes, whereby stronger neural oscillations were associated with longer-lasting percepts (*Figure 4A*, left; reproduced in *Figure 4— figure supplement 1A*, 'Duration' column). The 'Duration' neural subspace extracted from alpha and beta amplitudes contained significant predictive information for percept durations in the left-out test dataset, as evidenced by highly significant correlations between the actual percept duration and predicted percept duration according to neural data collected at different time points during a percept (*Figure 4A*, right, showing trial-by-trial correlation; *Figure 4—figure supplement 1B*, 'Duration'

column, showing predicted percept durations for trials with long vs. short actual percept durations). These results also show that neural activity related to perceptual stability is relatively persistent across time, evident from the beginning to the end of a percept. For SCP activity, significant predictive information over time was found for both images (*Figure 4—figure supplement 1B*, 'Duration' column); however, there was little consistency between $\beta$ weights across subjects (*Figure 4—figure supplement 1A*, 'Duration' column), or between the group-level $\beta$ weight maps for the two images (cosine similarity, $N = 272$ sensors, $\cos \theta = 0.08$, $p = 0.37$ assessed by a permutation test). By contrast, the group-level $\beta$ weight maps for the 'Duration' axis are highly correlated between the two images for alpha amplitude ($\cos \theta = 0.83$, $p = 0.02$) and beta amplitude ($\cos \theta = 0.82$, $p = 0.02$). Therefore, we conclude that information about perceptual stability, as captured by percept duration, is primarily carried by the amplitude of alpha- and beta-band activity. Although this information also exists in the SCP band, it is encoded in a less consistent manner across subjects and across different image inputs.

For the 'Switch' and 'Direction' axes, we found that all three neural features were significantly predictive of behavior (*Figure 4—figure supplement 1B*, 'Switch' and 'Direction' columns), suggesting that the neural representation of these processes is distributed across multiple frequency bands. For the 'Direction' axis, group-level $\beta$ weight maps for SCP and beta amplitude show significant sensors lateralized over the left hemisphere whose spatial distribution could be related to the button press response (carried out using the right hand). For the 'Switch' axis, the SCP-band $\beta$ weight maps are consistent with a dipole in the midline region corresponding to a potential source in the supplementary motor area; alpha/beta amplitudes have positive $\beta$ weights suggesting that the amplitudes decrease around the time of the perceptual switch, consistent with earlier findings (*de Jong et al., 2016*). Together, these findings provide a methodological validation of the present analysis approach; however, given the potential motor contribution to the results obtained from the 'Switch' and 'Direction' behavioral axes, we do not emphasize these findings henceforth.

## Role of neural oscillations in maintaining perceptual memory

Finally, we applied the state-space analysis to data from the *Discontinuous* condition, focusing on the blank periods (6 s each) between repeated image presentations (*Figure 1E*). For perceptual content reported before and after each blank period ('Pre' and 'Post' axes), we only found small temporal clusters of significant prediction in the test dataset for beta amplitude-defined neural subspace in the case of cube image (*Figure 4—figure supplement 2B*, 'Pre' and 'Post' columns). Overall, the information contained in neural activity during blank periods about perceptual content experienced earlier or later is weak, which is not surprising, given that there is no active perception related to the cube or face–vase image per se during this period.

However, all three neural features carried significant information about how far into the blank period the time point was (i.e., the temporal distance to previous image offset), suggesting a strong timing mechanism distributed across frequency bands. This is evident in the ability of the neural subspaces to predict timing information in the left-out test dataset (*Figure 4—figure supplement 2B*, 'Blank' column), as well as consistent sensor-level $\beta$ weight topography across subjects (*Figure 4—figure supplement 2A*, 'Blank' column). The SCP topography shows a midline dipole, and the alpha/beta topographies show widespread sensors whose oscillatory amplitudes decrease as time passes. Because the blank periods have a constant duration (6 s) before the next image onset, these results are consistent with previous reports of a contingent negative variation potential (CNV, an SCP activity with generators in the anterior cingulate cortex) and alpha amplitude decreases being neural correlates of temporal anticipation (*Nobre and van Ede, 2018*).

The most informative behavioral metric for the *Discontinuous* condition is the 'Memory' axis as it indicates the presence or absence of a perceptual memory trace (*Figure 3A*). Here, we found significant temporal clusters of prediction in the test dataset using alpha and beta amplitudes, but not SCP activity (which only showed small temporal clusters for the Cube image) (*Figure 4—figure supplement 2B*, 'Memory' column; alpha and beta results reproduced in *Figure 4B*). The consistency of the group-level $\beta$ weights across images was also stronger for alpha ($\cos \theta = 0.65$, $p = 0.08$) and beta amplitudes ($\cos \theta = 0.75$, $p = 0.10$) than for SCP activity ($\cos \theta = -0.11$, $p = 0.71$), although the consistency did not reach significance in any of the neural features. Interestingly, the group-level $\beta$ weight vectors are highly correlated between the Memory axis and the Blank axis for alpha and beta amplitudes (alpha amplitude: face–vase, $\cos \theta = 0.84$, $p < 0.01$; cube: $\cos \theta = 0.82$, $p = 0.01$; beta

amplitude: face–vase, cos $\theta$ = 0.83, p = 0.02, cube: cos $\theta$ = 0.86, p < 0.01), suggesting a strong timing component to the memory trace. The negative $\beta$ weights for the oscillation amplitudes (*Figure 4B*, left) show that alpha and beta oscillations are weaker when there is a perceptual memory trace.

Interestingly, the encoding of perceptual memory during the blank periods occurs first in beta activity (at ~0.5–1.5 s after blank onset), followed by alpha activity (at ~1.5–3 s), and is not significant in either frequency band during the latter half of the blank period (3–6 s) (*Figure 4B* and *Figure 4— figure supplement 2*). This transient encoding of perceptual memory in neural dynamics is consistent with a recent EEG study using a similar paradigm (*Zhu et al., 2022*). Speculatively, after the transient encoding in beta and alpha activities, perceptual memory trace might be maintained in short-term synaptic plasticity within the network in an 'activity-silent' state without measurable signatures in active neural dynamics (*Mongillo et al., 2008*; *Stokes et al., 2013*; *Rose et al., 2016*).

Together with the earlier results showing that stronger alpha/beta amplitudes promote perceptual stability (i.e., longer-lasting percepts) during continuous bistable perception (*Figure 4A*), these results show that the neural mechanisms supporting perceptual memory localize to the same frequency bands, but have different circuit-level mechanisms. In addition, sensors supporting perceptual memory (when the stimulus is temporarily removed from view) reside in more anterior regions than those supporting perceptual stability (when the stimulus is in view) (compare topoplots between *Figure 4A, B*), suggesting that higher-order brain circuits are recruited to maintain a perceptual memory trace when sensory input is absent, consistent with previous fMRI findings (*Wang et al., 2013*).

## Discussion

In this study, we dissected the roles that different types of neural activity play in perception. We found evidence that perceptual content is predominantly encoded in the SCP (<5 Hz) range, and no evidence of perceptual content encoding in the amplitude of alpha and beta oscillations. This was the case regardless of whether the sensory input is ambiguous or unambiguous. We additionally found that SCP activity along with the amplitude of alpha and beta oscillations encoded aspects of perceptual switching, including the distance to a switch and whether the current percept is stabilizing or destabilizing. However, information about how long the current percept would last and whether a perceptual memory trace would occur if the stimulus is temporally removed from view was primarily encoded in alpha and beta amplitudes. Together, these results show a frequency-band separation of information related to perceptual content and perceptual stability, with the former encoded in raw fluctuations of low-frequency SCP activity, and the latter primarily influenced by the amplitude fluctuations of alpha and beta oscillations.

Previous studies on bistable perception have typically focused on one aspect of perceptual behavior at a time, such as perceptual content or perceptual switching. By using a novel neural state-space analysis approach, we were able to simultaneously extract components of neural activity relevant to different aspects of perceptual behavior that all vary across time/trials and are mutually independent. Additionally, this approach can uncover important relationships between neural activity underlying different aspects of behavior. For example, for alpha and beta amplitudes, the state space extracted for 'Blank' and 'Memory' axes in the *Discontinuous* condition are strongly correlated, suggesting a strong timing mechanism to how perceptual memory is encoded during the blank period (i.e., the neural activity pattern associated with the presence of a perceptual memory trace is similar to the activity pattern that increases over time during the blank period). Compared to other multivariate analysis methods, the neural state-space method has specific advantages and is well suited to addressing the questions investigated herein. First, compared to multivariate decoding, the state-space method extracts multivariate neural activity patterns relevant to multiple behavioral metrics simultaneously, as opposed to investigating neural correlate of one behavioral metric at a time. Second, compared to automatic dimensionality reduction, such as PCA and similar techniques (*Churchland et al., 2012*; *Cunningham and Yu, 2014*; *Baria et al., 2017*), the state-space approach directly identifies the neural activity pattern (i.e., neural subspace) relevant to a particular behavioral metric, as opposed to being behavior agnostic.

Our finding of perceptual content encoding in the SCP band provides further evidence of the role of SCP in conscious perception, consistent with earlier studies (*Li et al., 2014*; *Baria et al., 2017*; *Flounders et al., 2019*). A general role of SCP in supporting conscious awareness (*He and Raichle, 2009*) is also corroborated by recent findings comparing different states of consciousness (*Bourdillon*

*et al., 2020*; *Toker et al., 2022*). In the domain of bistable perception, most studies probing neural correlates of perceptual content have employed fMRI (e.g., *Tong et al., 1998*; *Haynes and Rees, 2005*; *Wang et al., 2013*), and most electrophysiological studies have focused on changes in ERPs (e.g., *Britz et al., 2009*; *Pitts et al., 2009*), oscillatory power (e.g., *de Jong et al., 2016*), or neuronal firing rates (*Gelbard-Sagiv et al., 2018*) around perceptual switches. Previous electrophysiological studies probing neural correlates of perceptual content have typically used specialized stimulus design, such as frequency tagging (*Tononi et al., 1998*; *Srinivasan et al., 1999*), binocular rivalry involving face and oriented grating (where face-elicited ERFs, the M170, correlates with perceiving faces) (*Sandberg et al., 2013*; *Sandberg et al., 2014*), or auditory bistable stimuli where neural information integration correlates with perceiving an integrated auditory stream (*Canales-Johnson et al., 2020*). Here, by using classic ambiguous figures where the two percepts are symmetrical in salience and level of cortical processing, and showing results consistent across different images, our findings provide a more generalizable electrophysiological correlate of perceptual content. Our results also complement a recent intracranial electrophysiology study using the same ambiguous figures which revealed changes in corticocortical information flow depending on the specific perceptual content experienced (*Hardstone et al., 2021*). Finally, the potential role of gamma frequency band in encoding perceptual content should be further investigated in future studies using intracranial recordings which are more sensitive to gamma-band activity than MEG (e.g., *Panagiotaropoulos et al., 2012*).

A relationship between alpha and beta amplitudes and the stability of percepts has been reported in several previous studies of bistable perception (*Kloosterman et al., 2015*; *Piantoni et al., 2017*; *Zhu et al., 2022*). Although the detailed mechanisms involved remain unclear, two non-mutually exclusive mechanisms have been proposed: lateral inhibition and the resulting dynamical attractor at a local scale (*Piantoni et al., 2017*) and top-down feedback from higher-order regions (*Kloosterman et al., 2015*; *Zhu et al., 2022*). While both local inhibition and top-down processing roles have been ascribed to alpha and beta oscillations (*Jensen and Mazaheri, 2010*; *Michalareas et al., 2016*; *Spitzer and Haegens, 2017*), we believe that our finding of higher alpha/beta amplitude being associated with stronger perceptual stability (*Figure 4A*) is more compatible with a top-down interpretation. While lateral inhibition between competing neuronal groups is a key ingredient of biophysical models of bistable perception (e.g., *Shpiro et al., 2009*), and enhancing cortical inhibition by administering lorazepam, a GABA$_A$ receptor agonist, enhances perceptual stability (*van Loon et al., 2013*), lorazepam also has the effect of reducing alpha power (*Lozano-Soldevilla, 2018*)—opposite to the present finding of a positive correlation between perceptual stability and alpha power. By contrast, recent intracranial electrophysiological evidence suggests that top-down feedback can carry perceptual templates congruent with long-term priors that act to stabilize a particular percept (*Hardstone et al., 2021*). Given the well-documented role of alpha and beta oscillations in carrying top-down feedback (*van Kerkoerle et al., 2014*; *Bastos et al., 2015*; *Michalareas et al., 2016*), a plausible mechanism for the present finding of higher alpha/beta amplitudes being associated with longer percept durations is then a top-down modulatory influence carried in the alpha and beta bands.

Similarly, we interpret our finding of alpha and beta amplitudes being associated with perceptual memory as reflecting a top-down modulatory influence. Consistent with this interpretation, a previous fMRI study showed that the content of perception and perceptual memory during intermittent presentation of ambiguous images is especially decodable in higher-order frontoparietal regions, and that intermittent presentation elicits strong top-down influences as compared to continuous presentation (*Wang et al., 2013*). The locations of sensors involved in perceptual memory (*Figure 4B*) are more anterior than those involved in perceptual stability (*Figure 4A*), which may reflect the source and target of top-down modulation, respectively.

Our finding of alpha and beta amplitudes being related to perceptual memory is concordant with a recent EEG study (*Zhu et al., 2022*), but, superficially, the two studies appear to report opposite directions of this relationship: a negative correlation (manifested as negative $\beta$ weights) in the present study versus a positive correlation in the earlier EEG study. However, a closer inspection suggests that the two studies are in fact consistent: Zhu et al. used short blank durations (~0.5–1.5 s), and alpha/beta amplitudes are higher in stable-blank trials than unstable-blank trials early (within 500 ms of blank-onset) during the blank period, but lower in stable-blank trials (after 800 ms) late in the blank periods (*Figure 4B* therein). The present study used long blank durations (6 s), and the lower alpha/beta amplitude in stable-blank trials is mostly evident at 500 ms following blank onset or later

(*Figure 4B*, right, note 'projected average' is amplitude multiplied by $\beta$ weights, which are negative). The exact neurophysiological mechanisms contributing to these time courses remain to be investigated, but both studies converge to suggest that alpha and beta amplitudes influence not only perceptual stability when sensory input is in view but also perceptual memory when sensory input is temporarily removed from view. Importantly, the present results differ from previous studies showing beta power increases during working memory maintenance (*Spitzer and Haegens, 2017*), reinforcing the notion that perceptual memory differs from working memory: the former is unconscious and automatic (*Pearson and Brascamp, 2008*), while the latter is largely conscious and deliberate (*Trübutschek et al., 2019*). Furthermore, we found that neural activity (in the alpha and beta bands) underlying perceptual memory has significant overlap with neural activity encoding elapsed time (as evidenced by a significant positive correlation of $\beta$ weight vectors for the 'Blank' and 'Memory' axes), which also fits better with an automatic process as opposed to an working memory account (*Souza and Oberauer, 2015*; *Fulvio and Postle, 2020*).

In sum, across multiple perceptual conditions (unambiguous vs. ambiguous sensory input; continuous vs. intermittent presentation), we found that distinct components of dynamical neural activity contribute to the content vs. stability of perception. While perceptual content is encoded in the activity pattern of low-frequency neural activity in the SCP band, perceptual stability and perceptual memory are influenced by the fluctuations of alpha and beta oscillation amplitudes. These results provide clues to the neural mechanisms underlying stable visual experiences in the natural environment, wherein the ever-present noise and instability in the retinal images must be overcome to reconstruct the cause of sensory input in order to guide adaptive behavior. Finally, these results also inform future computational models of bistable visual perception and efforts to understand pathological processes underlying perceptual disorders in mental illnesses, including abnormal bistable perceptual dynamics in autism and schizophrenia (*Robertson et al., 2013*; *Kornmeier et al., 2017*; *Weinhammer et al., 2020*).

# Materials and methods
## Subjects
The experiment was approved by the Institutional Review Board of the National Institute of Neurological Disorders and Stroke (under protocol #14 N-0002). All subjects were right handed and neurologically healthy with normal or corrected-to-normal vision. Nineteen subjects between 19 and 33 years of age (mean age 24.5; nine females) participated in the MEG experiment. We excluded one subject from analysis due to repeatedly falling asleep during the task. All subjects provided written informed consent.

## Task design and behavioral analysis
The task was adapted from a previously run fMRI experiment (*Wang et al., 2013*). In the study, two well-known ambiguous images (Necker cube and Rubin face–vase) were used to study bistable perception under continuous (*Ambiguous* condition) and intermittent presentation (*Discontinuous* condition) (*Figure 1*). As a control, we also included a condition where we manipulated the content, outlines, and shading of the ambiguous images to accentuate one of the two percepts (*Unambiguous* condition), with the intention that the subject would perceive that percept.

Stimuli were presented using E-Prime Software (Psychology Software Tools, Sharpsburg, PA) via a Panasonic PT-D3500U projector with an ET-DLE400 lens, with the screen 55 cm from the subject's eyes. All face–vase images subtended 16.9 × 17.6 (height × width) degrees of visual angle, and all cube images subtended 14.3 × 14.5 degree.

Each subject completed 12 runs, consisting of 4 sets of 3 runs in the following order: Unambiguous, Ambiguous, and Discontinuous conditions.

Each Ambiguous run contained six trials, with each trial consisting of 2 s of written instruction, 2 s of fixation (while fixating on a crosshair in the center of the screen), 60 s of image presentation, and 3–7 s of intertrial interval (*Figure 1A*). Each ambiguous image was presented three times in a pseudo-random order. Subjects reported every spontaneous perceptual switch using their right hand via one of three buttons throughout the course of image presentation: one button for each of the possible percepts, and one for 'Unsure' which they were instructed to press if they experience neither or both

of the possible percepts. In order to investigate spontaneous perceptual switches, subjects were instructed to passively view the images and not to try to switch or hold onto a percept.

Each Unambiguous run contained 32 trials, with each block consisting of 2 s of written instruction, 2 s of fixation, 5 s of image presentation, and 3–7 s of intertrial interval (*Figure 1C*). The four unambiguous images were presented eight times each in a pseudorandom order. Subjects were asked to indicate their percept via one of three buttons (one button for each possible percept, and one for unsure) at each image presentation. Valid trials consisted of subjects pressing the button for the intended percept once, and no other button presses (*Figure 1D*).

Each Discontinuous run contained six trials, with each trial consisting of 2 s of written instruction, 2 s of fixation, nine repetitions of 2 s image presentation followed by a 6 s blank period (of which the last second contained the crosshair in the center of the screen) (*Figure 1E*). Subjects were asked to indicate their percept during each image presentation via a button press, and not to press buttons during the blank period. Perceptual switching during the 2 s image presentation was very rare and was excluded from analyses. The two ambiguous images were presented in alternating trials.

For all conditions, subjects were instructed to fixate upon a crosshair at the center of the screen at all times to avoid the potential influence of gaze on perception. Response mapping was altered between runs, by switching the buttons for the two percepts. For the first nine subjects the response mapping for the two percepts was switched after every run. For the final 10 subjects, we instead switched the response mapping after every set of 3 runs. Before entering the MEG, subjects performed practice runs until they were comfortable with the task and the buttons corresponding to each percept.

## MEG recordings

While performing the task, we recorded neural activity from each subject using a 275-channel whole-head MEG system (CTF). Three dysfunctional sensors were removed from all analyses. We also recorded gaze position and pupil size using a SR Research Eyelink 1000+ system. Eye-tracking was used for online monitoring of fixation and wakefulness during the experiment. MEG data were recorded at a sampling rate of 600 Hz, with a low-pass anti-aliasing filter of 150 Hz and no high-pass filter (i.e., DC recording). Before and after each run, the head position of a subject was measured using fiducial coils, in order to detect excessive movement. During each task subjects responded using a fibreoptic response button box. All MEG data samples were realigned with respect to the presentation delay of the projector (measured with a photodiode).

## MEG data preprocessing and feature extraction

All preprocessing and analysis of data were performed in MATLAB (Mathworks, Natick, MA) using custom-written code and the FieldTrip toolbox (*Oostenveld et al., 2011*). MEG data were first demeaned and detrended. Data were then filtered at 0.05–150 Hz using a third-order Butterworth filter, and line noise as well as harmonics were removed using fourth-order Butterworth band-stop filters (58–62, 118–122, and 178–182 Hz). Independent component analysis (Fieldtrip *runica* method) was then applied, and components were manually inspected to remove those related to eye blinks, eye movements, or heart-beat-related artifacts.

Three different features of neural activity were then extracted. SCP activity was obtained using a third-order low-pass Butterworth filter at 5 Hz. Alpha-band amplitude was extracted by taking the absolute of the Hilbert transform (Matlab, *abs(hilbert(data))*) of the preprocessed MEG data that had been filtered at 8–13 Hz using a third-order Butterworth filter. Beta-band amplitude was extracted in the same way, but using data filtered in the 13–30 Hz range.

## Decoding perceptual content

We attempted to decode perceptual content during the Ambiguous and Unambiguous conditions using the three extracted neural features (SCP, alpha amplitude, beta amplitude). For the *Ambiguous* condition, we first extracted periods between button presses where each button press was for a different percept (excluding 'Unsure' button presses), and the period was labeled according to the first button press (i.e., Face, Vase, Green, or Blue). As these periods were all of different durations, we then rescaled them to be the same length by selecting 100 equally spaced time points, giving us percentiles of the percept's duration. For the *Unambiguous* condition, we selected valid trials for analysis, wherein the subject only pressed a button once for the intended percept. The time period

used for decoding was the 5 s that the image was on the screen, and MEG data were downsampled to 10 Hz before applying the decoding pipeline. The label of the trial was the image that was presented (i.e., Face, Vase, Green, or Blue). For both task conditions, the classification was done separately for the Necker Cube (Green vs. Blue) and Rubin face–vase (Face vs. vase). All trials were normalized (z-scored) across sensors at each time point. Trials were then split into fourfolds with an equal number of trials of each label in each fold. Trials for a fold were selected by taking every fourth trial of that trial type (ordered by when the trial occurred during the recording).

The decoding pipeline consisted of taking one fold as the testing set, and training a linear SVM classifier (cost = 1) using the LIBSVM packages (*Chang and Lin, 2011*) at each time point to the trials from the other threefolds, which constituted the training set. Decoding accuracy of the classifier was then calculated on the testing set. The temporal cross-generalization of the classifier was also tested by assessing its classification accuracy at every other time point. This was done using each fold as the testing set, and the decoding accuracy (and temporal generalization) was averaged across the fourfolds.

## Cluster-based permutation tests for multivariate pattern decoding

The group-level statistical significance of classifier accuracy at each time point was assessed by a one-tailed, one-sample Wilcoxon signed-rank test against chance level (50%). To correct for multiple comparisons, we used cluster-based permutation tests (*Maris and Oostenveld, 2007*). Temporal clusters were defined as contiguous time points with above-threshold classification accuracy (cluster-defining threshold: $p < 0.1$). The test statistic $W$ of the Wilcoxon signed-rank test was summed across time points in a cluster to yield a cluster's summary statistic. Cluster summary statistics were compared to a null distribution, constructed by shuffling class labels 100 times, and extracting the largest cluster summary statistic for each permutation. Clusters in original data with summary statistics exceeding the 95th percentile of null distribution were considered significant (corresponding to $p < 0.05$, cluster-corrected, one-tailed test). For classifier temporal generalization, the permutation-based approach for cluster-level statistical inference used the same procedure as above, where clusters were defined as contiguous time points in training and/or generalization dimensions with above threshold ($p < 0.1$) classification accuracy.

## Neural state-space analysis

To work out the relative contributions of different behaviors to neural activity patterns, we developed a novel multivariate analysis method to extract the neural subspace relevant to each behavior, following the approach used in *Mante et al., 2013*. While perceptual content is clearly an important aspect of behavior, there are other aspects of behavior which account for the perceptual switching dynamics (*Ambiguous* condition) and perceptual memory (*Discontinuous* condition). For the *Ambiguous* condition, we first selected 100 equally spaced time points from each period that occurred between button presses for the two percepts (i.e., not for time points preceded or followed by an unsure button press). We then defined four behavioral metrics for each time point:

- *Type*, a binary variable indicating the current percept.
- *Duration*, a continuous variable which takes the same value throughout a percept and is normalized within subject (i.e., 0 for the shortest percept reported and 1 for the longest percept).
- *Switch*, a continuous variable that was 0 at the time of a button press and 1 at the midway point between button presses, indicating the relative temporal distance to perceptual switches.
- *Direction*, a binary variable indicating whether the current percept is stabilizing (i.e., time point is in the first half of its duration) or destabilizing (i.e., in the second half of its duration).

For the *Discontinuous* condition, only time points during the blank period (6 s total, including the 1 s fixation period) were used, where the blank period was preceded and followed by an image presentation during which the subject pressed for one of the two percepts. Four behavioral metrics were defined.

- *Pre*, a binary variable indicating the percept reported before the blank period.
- *Post*, a binary variable indicating the percept reported after the blank period.
- *Blank*, a continuous variable increasing from 0 at the beginning of the blank period to 1 at the end of the blank period, indicating time elapsed during the blank period.

- *Memory*, a binary variable indicating whether the percept before the blank was the same as that after the blank, with 1 indicating the presence of a memory trace and 0 indicating the absence of a memory trace.

These time points were split into two datasets (first and second half of time points based on time through experiment), with the first half used as the training set, and the second half used as the test set. Using the training dataset, the MEG data (applied separately for the three neural features: SCP, alpha amplitude, and beta amplitude) for each sensor are first normalized over time using the mean and standard deviation from the training set. A multilinear regression of the following form was solved to find the $\beta$ weights in the equation (using MATLAB function *fitlm*, with *RobustOpts*).

$$MEG_{sensor1} = \beta_{a1}Behavior_a + \beta_{b1}Behavior_b + \beta_{c1}Behavior_c + \beta_{d1}Behavior_d + \varepsilon$$

These $\beta$ weights define the axes of the behavioral subspace for that neural feature. MEG data from the test set (which has been normalized at the individual sensor level using the mean and standard deviation from the training set, so that the projection method does not depend on any information from the test set) can then be projected onto the behavioral axes (using the MATLAB function *mldivide*). This gives a prediction of the value for each behavioral metric based on neural activity at each time point of the test set.

## Statistics for neural state-space analysis

To assess the consistency across subjects of the $\beta$ weights defining each behavioral axis, and whether each neural feature carried predictive information about the behavior, a cluster-based permutation method (*Maris and Oostenveld, 2007*) was applied separately for each axis. This involved shuffling the behavioral information across all of the trials only for that axis. For axes where behavior was defined in the same way for each trial (Ambiguous: Switch and Direction Axes; Discontinuous: Blank Axis), each trial was instead 'flipped' with a 50% chance, where the flipped trial was equal to 1 — original behavior. Once the behavioral data were shuffled, the state-space analysis was reapplied, and this was done for 100 permutations of the data.

To assess consistency of the $\beta$ weights defining each behavioral axis, a two-sided Wilcoxon sign-rank test against zero was applied separately to each electrode. To correct for multiple comparisons across sensors, we used cluster-based permutation tests (*Maris and Oostenveld, 2007*). Spatially contiguous electrodes with a significant bias ($p < 0.05$) and the same sign of the test statistic $W$, formed a cluster (either positive or negative depending on the sign) where the cluster summary statistic was the sum of the electrodes' test statistic $W$. This cluster summary statistic was compared to a null distribution, formed from the largest (i.e., most positive or most negative) cluster summary statistic for each permutation. Clusters in original data with summary statistics exceeding the 95th percentile of null distribution were considered significant (corresponding to $p < 0.05$, cluster-corrected, one-tailed test). Positive and negative clusters were assessed separately by comparison to the respective null distribution.

To assess consistency between two state spaces (either between $\beta$ weight maps for the two different images and the same axis, or between two different axes and the same image), cosine similarity was applied.

$$\cos\theta = \frac{\boldsymbol{\beta_A} \cdot \boldsymbol{\beta_B}}{\|\boldsymbol{\beta_A}\|\|\boldsymbol{\beta_B}\|}$$

where $\boldsymbol{\beta_A}$ and $\boldsymbol{\beta_B}$ correspond to the two $\beta$ weight maps. Cosine similarity produces a value between −1 and 1, where 1 indicates that the projection of the MEG data into the two behavioral state spaces would produce perfectly correlated behavioral estimates, and −1 indicates that they would be entirely anticorrelated. To assess significance of the cosine similarity, a null distribution was created by calculating cosine similarity between the two $\beta$ maps for each permutation, and the p value was calculated as the fraction of this distribution with values larger than the cosine similarity of the original data.

To assess whether each neural feature carried predictive information about each behavioral axis, a different test was applied depending on the axis. In the following analyses, a one-sided statistical test is often used because if the neural subspace is successfully extracted, the predicted behavioral metric in the test dataset should follow specific relationships similar to the definition of these axes (*Figure 3A*).

First, data from the test dataset were projected into the neural subspace defined for each axis using the training data. For those axes with binary behavior that had the same value across time for each trial (Ambiguous: Type; Discontinuous: Pre, Post), a paired *t*-test across subjects (one-sided) was applied at each time point between the trial-averaged predicted behavior for the two groups of trials (e.g., Green Cube and Blue Cube).

For the axis with continuous behavior that had the same value across time for each trial (Ambiguous: Duration), two analyses were carried out: (1) (*Figure 4A*) For each subject, predicted percept duration (using neural activity at each time point) was correlated with the actual percept duration across trials by a Spearman correlation. The rho values were then subjected to a Wilcoxon sign-rank test across subjects (one-sided), and corrected for multiple comparisons using cluster-based permutation test.

A one-sided Wilcoxon sign-rank test was applied at each time point between the predicted and actual behavior. (2) (*Figure 4—figure supplement 1B*) A median split was performed on the test dataset for each subject according to percept duration. The projected data (i.e., predicted percept duration based on neural data) were then compared between the two groups of trials by a paired *t*-test across subjects (one-sided), and corrected for multiple comparisons using cluster-based permutation test.

For those axes where the actual behavior changed across time within a trial in a continuous manner (Ambiguous: Switch; Discontinuous: Blank), the trial-averaged predicted behavior from each subject was compared to the actual behavior using a Spearman correlation. The Spearman rho values were then subjected to a group-level test using a Wilcoxon signed-rank test against zero (one-sided).

Lastly, for the axis where actual behavior changed across a trial in a binary manner (Ambiguous: Direction), the trial-averaged predicted behavior for each subject was compared between the first half of the trial and the second half of the trial using a two-sample *t*-test. The resulting *t*-values were then subjected to a Wilcoxon signed-rank test against zero (one-sided) at the group level.

To correct for multiple comparisons, we used cluster-based permutation tests (*Maris and Oostenveld, 2007*). Temporal clusters were defined as contiguous time points with above-threshold test statistic (cluster-defining threshold: $p < 0.05$). The test statistic (Wilcoxon signed rank (*W*) or *t*-test (*t*)) was summed across time points in a cluster to yield a cluster's summary statistic. Cluster summary statistics were compared to a null distribution, formed by extracting the largest cluster summary statistic for each permutation. Clusters in original data with summary statistics exceeding the 95th percentile of null distribution were considered significant (corresponding to $p < 0.05$, cluster-corrected, one-tailed test).

## Acknowledgements

This work was supported by a National Science Foundation CAREER Award (BCS-1753218), an Irma T Hirschl Career Scientist Award, and a National Institutes of Health grant (R01EY032085) to B.J.H. M.Z. was supported by NIH Training Grant R90DA043849.

## Additional information

### Funding

| Funder | Grant reference number | Author |
|---|---|---|
| National Science Foundation | BCS-1753218 | Biyu J He |
| Irma T. Hirschl Trust | | Biyu J He |
| National Institutes of Health | R01EY032085 | Biyu J He |

The funders had no role in study design, data collection, and interpretation, or the decision to submit the work for publication.

## Author contributions
Richard Hardstone, Formal analysis, Validation, Investigation, Visualization, Methodology, Writing – original draft, Writing – review and editing; Matthew W Flounders, Michael Zhu, Investigation; Biyu J He, Conceptualization, Resources, Supervision, Funding acquisition, Validation, Writing – original draft, Project administration, Writing – review and editing

## Author ORCIDs
Richard Hardstone  http://orcid.org/0000-0002-7502-9145
Matthew W Flounders  http://orcid.org/0000-0001-7014-4665
Biyu J He  http://orcid.org/0000-0003-1549-1351

## Ethics
This experiment was approved by the Institutional Review Board of the National Institute of Neurological Disorders and Stroke (under protocol #14 N-0002). All subjects provided written informed consent for the research use and eventual publication of their data.

## Decision letter and Author response
Decision letter https://doi.org/10.7554/eLife.78108.sa1
Author response https://doi.org/10.7554/eLife.78108.sa2

# Additional files

## Supplementary files
• MDAR checklist

## Data availability
The dataset generated by this study, including data and code to reproduce all the figures, are shared through figshare: doi:https://doi.org/10.6084/m9.figshare.19375910.

The following dataset was generated:

| Author(s) | Year | Dataset title | Dataset URL | Database and Identifier |
|---|---|---|---|---|
| Hardstone R, Flounders MW, Zhu M | 2022 | Analysis Scripts, Plotting Scripts, Data for Plotting | https://figshare.com/s/c328299bea96686c2eec | figshare, 10.6084/m9.figshare.19375910 |

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
