## [Editor Report]

Bistable visual perception offers a unique window to study how perception arises and changes via an interaction between bottom-up and top-down processes. In three Magnetoencephalography (MEG) experiments with advanced neural state space analysis, this study demonstrates that two key aspects of bistable visual perception – perceptual content and perceptual stability – are mediated by slow cortical potential (SCP) and α-β-band neural oscillations, respectively. The findings will be of interest for many fields, including those studying perception, consciousness, and attention.

---

## [Decision Letter]

**Decision letter after peer review:**

Thank you for submitting your article "Frequency-specific neural signatures of perceptual content and perceptual stability" for consideration by *eLife*. Your article has been reviewed by 2 peer reviewers, one of whom is a member of our Board of Reviewing Editors, and the evaluation has been overseen by Chris Baker as the Senior Editor. The reviewers have opted to remain anonymous.

We are very sorry for taking so long to make our decisions. We generally need three reviewers but now go ahead with the current two received reviews to avoid further delays. Both reviewers acknowledged your important findings but also raised several concerns that need substantial revision and new results. There are also other suggestions for you to consider.

Essential revisions:

1) Need more results supporting the dissociation of perceptual content and perceptual stability as claimed in the paper. For example, the statistical evidence for the specificity of SCP to perceptual content (1st point by Reviewer 2), the neural representation of perceptual duration also in SCP in addition to α-β oscillation (1st point by Reviewer 1), the exact relationship between SCP and α-β oscillation (2nd point by Reviewer 1), and the vague results about the α-β tracking of memory under discontinuous condition (3rd point by Reviewer 1).

2) Reviewer 2 raised substantial concerns about the rationale of the behavioral metrics used in the neural state space analysis (see 2nd point by Reviewer 2), considering the power-law distribution of perceptual duration during bistable perception. This is a critical point since it would challenge the core assumption of the main analysis. The authors should provide strong evidence to verify their analysis rationale.

*Reviewer #1 (Recommendations for the authors):*

(1) The main conclusion of the paper is the dissociation of SCP and α-β oscillation for perceptual content and perceptual stability, respectively. Meanwhile, the results for perceptual stability using the neural state-space analysis could not fully support the dissociation claim. As shown in Figure S2, the SCP showed clear representations of perceptual duration, similar to that for α and β power. The author argued that the corresponding spatial map for SCP is less reliable compared to α-β, but I am not convinced that this would serve as strong evidence excluding SCP's role.

(2) Related to the above point, I would suggest the authors dig into the exact relationship between SCP and α-β oscillation, particularly regarding their functions in perceptual stability. For example, α and β might be the key top-down modulation signal to sustain the perception, which would contribute to the observed perceptual duration effect in SCP.

(3) As shown in Figure 4, the authors stated that the α-β tracked the memory during the blank interval for the Discontinuous condition, while SCP could not. Meanwhile, the results look vague and noisy. Instead of comparing stable and unstable trials, I am confused that why the authors do not plot a single time course denoting the memory tracking instead of the current two lines (stable and unstable). Moreover, the difference between stable and unstable seems to occur at random times and is not stable between conditions. How to interpret these inconsistencies? Finally, the SCP displayed a similar (stable>unstable) trend and the cube condition even showed a significant difference. I think there should be consistent criteria to define whether or not there is significant tracking.

*Reviewer #2 (Recommendations for the authors):*

It is not necessary but I just wonder what would happen if we use γ band data. If the authors have some solid reasons to exclude the band, I hope such reasons would be clarified in the text.

---

## [Author Response]

Essential revisions:1) Need more results supporting the dissociation of perceptual content and perceptual stability as claimed in the paper. For example, the statistical evidence for the specificity of SCP to perceptual content (1st point by Reviewer 2), the neural representation of perceptual duration also in SCP in addition to α-β oscillation (1st point by Reviewer 1), the exact relationship between SCP and α-β oscillation (2nd point by Reviewer 1), and the vague results about the α-β tracking of memory under discontinuous condition (3rd point by Reviewer 1).

We have now included additional results regarding the specificity of SCP to perceptual content (new Figure 2—figure supplement 1), additional discussion on the role of SCP in perceptual duration (and its relationship to the role of α and β oscillations, see Author response image 1), and have replotted the α and β tracking of perceptual memory under discontinuous condition which shows highly reliable results between the two images (Figure 4). Detailed responses are included below.

2) Reviewer 2 raised substantial concerns about the rationale of the behavioral metrics used in the neural state space analysis (see 2nd point by Reviewer 2), considering the power-law distribution of perceptual duration during bistable perception. This is a critical point since it would challenge the core assumption of the main analysis. The authors should provide strong evidence to verify their analysis rationale.

See our response to reviewer 2 below, which (1) disputes that the distribution of percept durations follows a power law (it is a γ distribution); and (2) includes a control analysis which shows that the exact choice of the behavioral timing function is not critical for the presented results.

Reviewer #1 (Recommendations for the authors):(1) The main conclusion of the paper is the dissociation of SCP and α-β oscillation for perceptual content and perceptual stability, respectively. Meanwhile, the results for perceptual stability using the neural state-space analysis could not fully support the dissociation claim. As shown in Figure S2, the SCP showed clear representations of perceptual duration, similar to that for α and β power. The author argued that the corresponding spatial map for SCP is less reliable compared to α-β, but I am not convinced that this would serve as strong evidence excluding SCP's role.

We have now toned down the claim of separation of neural activity for perceptual stability, and have highlighted that SCP carries information about perceptual stability.

Abstract:

“Perceptual stability is additionally influenced by the amplitude of α and β oscillations.”

Introduction:

“We therefore hypothesized that there might exist a frequency-band separation between neural activity supporting perceptual content and neural activity supporting perceptual stability, with the former residing in the non-oscillatory activity in the SCP range, and the latter predominantly residing in oscillatory activity in the α/β range.”

“Surprisingly, we also found that SCP modulated perceptual stability, although with less spatial consistency across subjects compared to α and β oscillations.”

We believe that the significance of the spatial map is informative about the strength of the role of a neural feature in the behavior. The spatial map represents multivariate regressions, and its weights can be interpreted as evidence for the involvement of a neural feature in a behavior (unlike decoder weight maps where it is unclear if a strong weight is due to signal or suppressing noise see Haufe et al., Neuroimage 2014). Further, we show that there is no consistency between the two images in how percept duration is encoded in SCP (as reflected in their different spatial topography), suggesting that the relation of SCP activity to perceptual duration is image-specific.

Therefore, while SCP does carry information about perceptual stability, it likely does so in a more distributed and less anatomically defined way as compared to α and β oscillations where the information is predominantly in posterior regions. In addition, the image-specific relationship between SCP and percept duration suggests that it is not picking up on a generic perceptual stability mechanism. We have added a sentence to the relevant *Results section* to clarify this point:

“Therefore, we conclude that information about perceptual stability, as captured by percept duration, is primarily carried by the amplitude of α- and β-band activity. Although this information also exists in the SCP band, it is encoded in a less consistent manner across subjects and across different image inputs.”

In sum, we hope that revisions to the text mentioned above address the concern raised by the reviewer by clearly stating that SCP also modulates perceptual stability.

(2) Related to the above point, I would suggest the authors dig into the exact relationship between SCP and α-β oscillation, particularly regarding their functions in perceptual stability. For example, α and β might be the key top-down modulation signal to sustain the perception, which would contribute to the observed perceptual duration effect in SCP.

To test the relationship between SCP and α-β oscillation in their functions in perceptual stability, we compared the neural activity projected into the sub-spaces for percept duration for the three neural features (SCP, α, β). We did this by applying partial correlation between each projected neural feature and the behavior, regressing out the influence of the other two neural features. We then compared this partial correlation with the original correlation. For SCP we found no significant reduction in the correlation (Author response image 1, left column), suggesting that SCP carries different information about perceptual stability compared to α and β oscillations. For α and β we did find a significant reduction (Author response image 1, middle and right columns), suggesting that α and β oscillations exert partially redundant influences on perceptual stability.

Since this analysis does not directly speak to the reviewer’s hypothesis about α and β being involved in top-down modulation and is somewhat peripheral to our main analyses, we have elected not to include it in the revised manuscript.

**Author response image 1. sa2fig1:** SCP carries unique information about perceptual stability. Black trace shows reduction in correlation with percept duration when regressing out the other two neural features; shaded areas show s.e.m. across subjects. Red bars indicate time points where partial correlation was significantly reduced as compared to the original correlation (one-sided t-test, *p*<0.05, cluster-based permutation test with 1000 permutations).

(3) As shown in Figure 4, the authors stated that the α-β tracked the memory during the blank interval for the Discontinuous condition, while SCP could not. Meanwhile, the results look vague and noisy. Instead of comparing stable and unstable trials, I am confused that why the authors do not plot a single time course denoting the memory tracking instead of the current two lines (stable and unstable). Moreover, the difference between stable and unstable seems to occur at random times and is not stable between conditions. How to interpret these inconsistencies? Finally, the SCP displayed a similar (stable>unstable) trend and the cube condition even showed a significant difference. I think there should be consistent criteria to define whether or not there is significant tracking.

We thank the reviewer for this suggestion, and agree that plotting the difference between stable and unstable trials is more intuitive. The new plots (included in the revised Figure 4 and Figure 4—figure supplement 2) emphasize that for β oscillations there is a consistent relationship with perceptual memory at ~0.5–1.5 seconds, and for α oscillations there is a consistent relationship with perceptual memory at ~1.5–3 seconds. By contrast, SCP’s relationship with perceptual memory occurs later, more transiently, and is only significant for one of the two ambiguous images tested.

We do not think that the neural correlate for perceptual memory should necessarily manifest in the neural dynamics in a stable fashion across the blank period, even though, behaviorally, perceptual memory lasts throughout this period. As is now well established in the working memory literature (e.g., Stokes et al., Neuron 2013; Murray et al. PNAS 2017), working memory traces that last across a long delay period are supported by heterogeneous and temporally changing neural dynamics, including ‘activity-silent states” that would not manifest in population activity recordings such as MEG. Finally, our finding of transient neural activity correlate of perceptual memory is also consistent with a recent EEG study using a similar task (Zhu et al., Sci Rep 2022). We have now included these considerations in the relevant Results section:

“Interestingly, the encoding of perceptual memory during the blank periods occurs first in β activity (at ~0.5–1.5 sec after blank onset), followed by α activity (at ~1.5–3 sec), and is not significant in either frequency band during the latter half of the blank period (3–6 sec) (Figure 4B and Figure 4—figure supplement 2). This transient encoding of perceptual memory in neural dynamics is consistent with a recent EEG study using a similar paradigm (Zhu et al., 2022). Speculatively, after the transient encoding in β and α activity, perceptual memory trace might be maintained in short-term synaptic plasticity within the network in an “activity-silent” state without measurable signatures in active neural dynamics (Mongillo et al., 2008; Stokes et al., 2013; Rose et al., 2016).”

Finally, regarding the criteria for deciding whether there is significant neural tracking of a behavior, one criterion that we rely quite heavily on throughout the manuscript is the consistent and reproducible findings between the two ambiguous images. Although not mandated by the current norm in the field, we felt strongly that collecting data using two separate ambiguous images and separately analyzing data from them allowed us to perform a within-study reproducibility and generalizability check. Therefore, the results that are consistent between images are not only reproduced (using independent data from the same subjects) but also generalize across the specific choice of stimulus. Our lab has adopted this strategy to ensure that our results are robust and stand the test of time in several previous publications on bistable perception (Wang et al., PNAS 2013; Hardstone et al., Nat. Commun. 2021; Zhu et al., Sci Rep 2022). Because SCP’s relationship with perceptual memory fails this important criterion, we do not emphasize it in the conclusions.

Reviewer #2 (Recommendations for the authors):It is not necessary but I just wonder what would happen if we use γ band data. If the authors have some solid reasons to exclude the band, I hope such reasons would be clarified in the text.

In initial analysis we did not find significant decoding of perceptual content using γ-band power, consistent with our earlier MEG study (Baria et al., PLoS Comput Biol 2017). While it is likely that γ band does carry perceptual coding information (e.g., see Panagiotaropoulos et al., Neuron 2012), we believe that this question would be best addressed using intracranial recordings where SNR for γ-band activity is better. We have added a sentence in Discussion to this point:

“Finally, the potential role of γ frequency band in encoding perceptual content should be further investigated in future studies using intracranial recordings which are more sensitive to γ-band activity than MEG [e.g., (Panagiotaropoulos et al., 2012)].”